# Rearing methods and life cycle characteristics of *Chironomus* sp. *Florida* (Chironomidae: Diptera): A rapid-developing species for laboratory studies

**Roberto Reyes-Maldonado**[1,2]*, **Bruno Marie**[2,3], **Alonso Ramírez**[4]

**1** Department of Biology, University of Puerto Rico, San Juan, Puerto Rico, **2** Institute of Neurobiology, University of Puerto Rico, San Juan, Puerto Rico, **3** Department of Anatomy & Neurobiology, University of Puerto Rico, San Juan, Puerto Rico, **4** Department of Applied Ecology, North Carolina State University, Raleigh, North Carolina, United States of America

* roberto.reyes3@upr.edu, robertoomaldo@gmail.com

**Data Availability Statement:** All relevant data are within the manuscript and its Supporting Information files.

## Abstract

The species *Chironomus* sp. *"Florida"* has several qualities that make it a potential aquatic laboratory model to be used in Puerto Rico. Its use as such, however, requires a rearing protocol and life cycle description not previously reported. The present study addresses this lack of information by first describing a rearing method obtained through three years of observations. Next we describe and discuss the life cycle and the effects of temperature and feeding on development. The species has a short life cycle (typically 11 days) and larval stages easily identified using body measurements. Temperature affects the duration of the life cycle, with warm temperatures producing faster development than cold temperatures. The effects of different food concentrations vary: in large water volumes, concentrations of 2 mg/larva/day produce faster developmental times, but at low water volumes, small food concentrations of 0.5 mg/larva/day produce faster developmental times. The rearing protocol and life cycle parameters presented in this study are intended to promote the use of this species as a laboratory model. The fast development of *Chironomus* sp. *"Florida"* makes it ideal for toxicological studies.

## Introduction

Chironomids are one of the most common insects inhabiting freshwater bodies. The larvae are ubiquitous and can be found in any aquatic environment, from lakes to phytotelmata [1]. A great part of the species in this group are benthonic, some of them feeding on sediment, detritus, and other types of organic matter [2]. Larval morphological characteristics are diverse, and some groups are noticeable by their red colour, which is why many of the species are commonly called bloodworms. This colouration is given by a type of haemoglobin, a protein that is used for fixing oxygen and possibly metabolizing some environmental contaminants [3]. This

**Funding:** This work was supported by the Puerto Rico Centre for Environmental Neuroscience (PRCEN) under NSF-Grant HRD 1736019 and by the NIH NIGMS GM103642 grant to BM.

**Competing interests:** The authors have declared that no competing interests exist.

characteristic of chironomids and their symbiotic relation with endogenous bacterial communities give them the advantage of tolerating and colonizing polluted environments [4, 5].

Despite the larval tolerance to contaminated environments, chironomids are prone to exhibit changes and disruptions at molecular, morphological, and behavioural levels as a response to their interaction with contaminants [6, 7]. For this reason, scientists have used them as freshwater bioindicators and laboratory models, taking their tolerance to infer ecosystem conditions and their responses to assess toxicity or chemical interactions. For example, their relative abundance is used in biotic indices (e.g., [8, 9]); mouth part deformities are used as a markers for toxicity (e.g., [10, 11]); and species survivorship are often used to calculate LC50 values for chemicals (e.g., [12]). The genus *Chironomus* has been used as a laboratory model, in particular *C. riparius* and *C. tentans*. These species are ecologically important and are easy to culture under laboratory conditions [13]. They have been used principally in temperate zones but studies in tropical regions (e.g., Brazil, northern Argentina, and India) recommend the use of local or tropical species, even though there is little information on them [14–16]. Using local species as models might result in a better alternative as they are adapted to local environmental conditions [17].

In the Insular Caribbean, native chironomid species have been used as indicators to infer stream conditions, but not as laboratory models. Researchers have focused on the use of exotic aquatic animals as model organisms for assessing chemical toxicity (e.g., *Danio rerio*, [18, 19]). The information collected from them, however, does not have the appropriate ecological relevance that native species have. Studies have reported using the native species *Macrobrachium carcinus* to assess the effects of chemical pollution [20], but this organism's complex life cycle complicates its maintenance and reproduction under laboratory conditions. The use of a *Chironomus* species could solve these problems since chironomids are easy to maintain and can be found in all freshwater bodies in the Insular Caribbean region [21–24].

In Puerto Rico, we have been studying a native *Chironomus* species to determine its use as a laboratory model for assessing toxicity and responses to chemical pollution. The species *Chironomus* sp. "*Florida*" (as provisionally called by Epler [25]) has several qualities that make it a potential laboratory model. Indeed, it has a widespread distribution through the central and north-eastern part of the island, its larvae can be found inhabiting the sediments of streams and temporary pools at any time of the year, and all of its stages can be maintained under laboratory conditions. Nevertheless, no rearing protocol and life cycle data have yet been reported to recommend this species as a laboratory model. The present study addresses this lack of information by describing the rearing method that we have developed for *Chironomus* sp. "*Florida*" through 3 years of observations. In addition, we provide details on the life cycle of the species, presenting some effects observed when animals are exposed to different temperatures and food concentrations.

## Materials and methods

### Field collection and species identification

We initiated all cultures by collecting egg masses in the field, thus securing hundreds of larvae of the same species without major effort. The best method for securing egg masses of *Chironomus* sp. "*Florida*" in Puerto Rico was to place water-filled dark containers in open areas. The water in these containers reflected polarized light, which gravid females used as cues for selecting places to lay eggs [26]. Eggs masses were easy to observe on the water's surface, attached to the edge or floating, during the evening and morning 1 to 7 days after placing the containers. They were collected by suctioning with a Pasteur pipette or by detaching and lifting them with

the tip of the finger or a small wooden stick. Easily identifiable, this was the only species on the island with straight flat-shaped egg masses (Fig 1).

A residential area in San Juan (18˚24'5.02"N, 66˚3'11.47"W, 625 meters away from Río Piedras mainstem) was used successfully since 2017 to obtain egg masses with this method. Egg masses of *Chironomus* sp. *"Florida"* also were collected in a more time-consuming way by sweep-netting emerged aquatic plants at Quebrada San Antón, Carolina (18˚25'04.7"N 66˚00'03.0"W), and by direct collection in temporary pools, artificial channels, and animal watering troughs at Río Bauta, Morovis (18˚15'41.4"N 66˚27'25.9"W) and Quebrada Buruquena, Río Grande (18˚19'17.7"N 65˚49'10.9"W).

Egg masses were placed in clean Petri dishes with dechlorinated tap water until hatching. A single species culture of *Chironomus* sp. *"Florida"* was started by placing the larvae from five to ten masses into a single culture aquarium, as described below. Species identity was confirmed at the larval stage using cues described in Epler [25].

## Culture apparatus

Our *Chironomus* sp. *"Florida"* rearing apparatus consisted of a series of interconnected aquariums with an overflow system (Fig 2). Aquariums were made of plastic, with a capacity of 8 L (29x16x18 cm) (Fig 2.1), and an aerial enclosure (15.5 cm height) made of two curved wires (50 cm length) intersected and attached at each corner top of the aquarium (Fig 2.2). Each

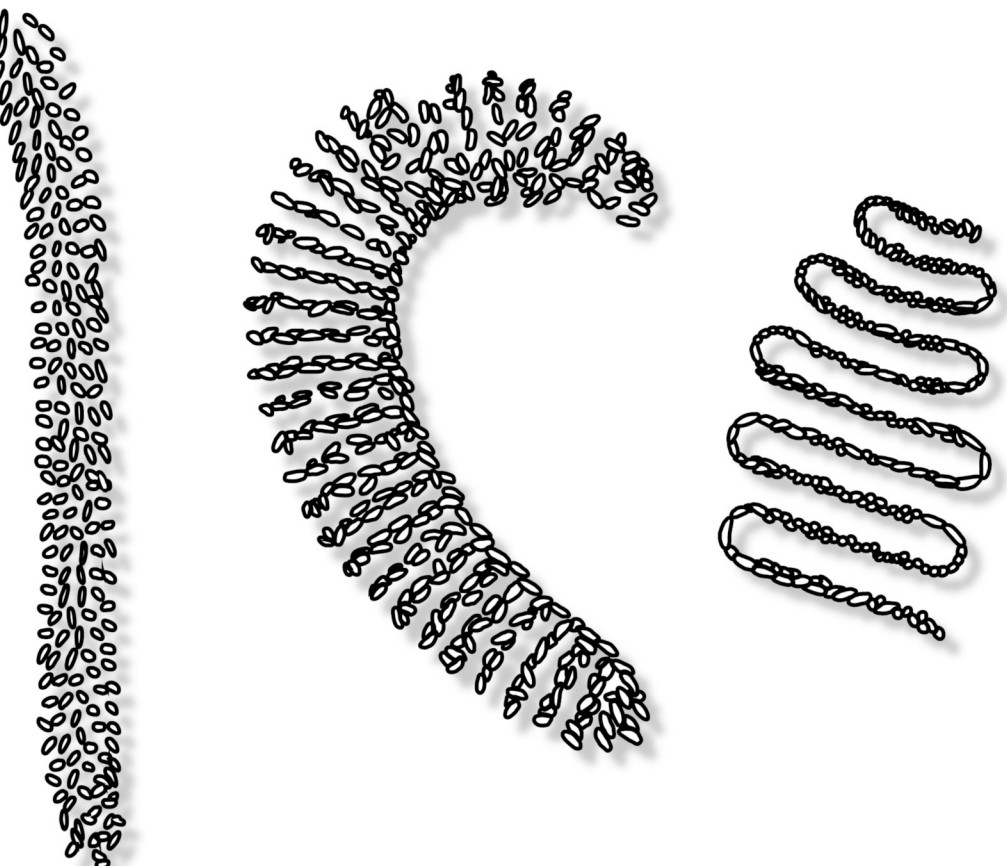

**Fig 1. Morphological diversity of chironomid egg masses collected in the field.** (Left) *Chironomus* sp. "*Florida*". (Centre) *Chironomus* sp. (Right) *Dicrotendipes* sp.

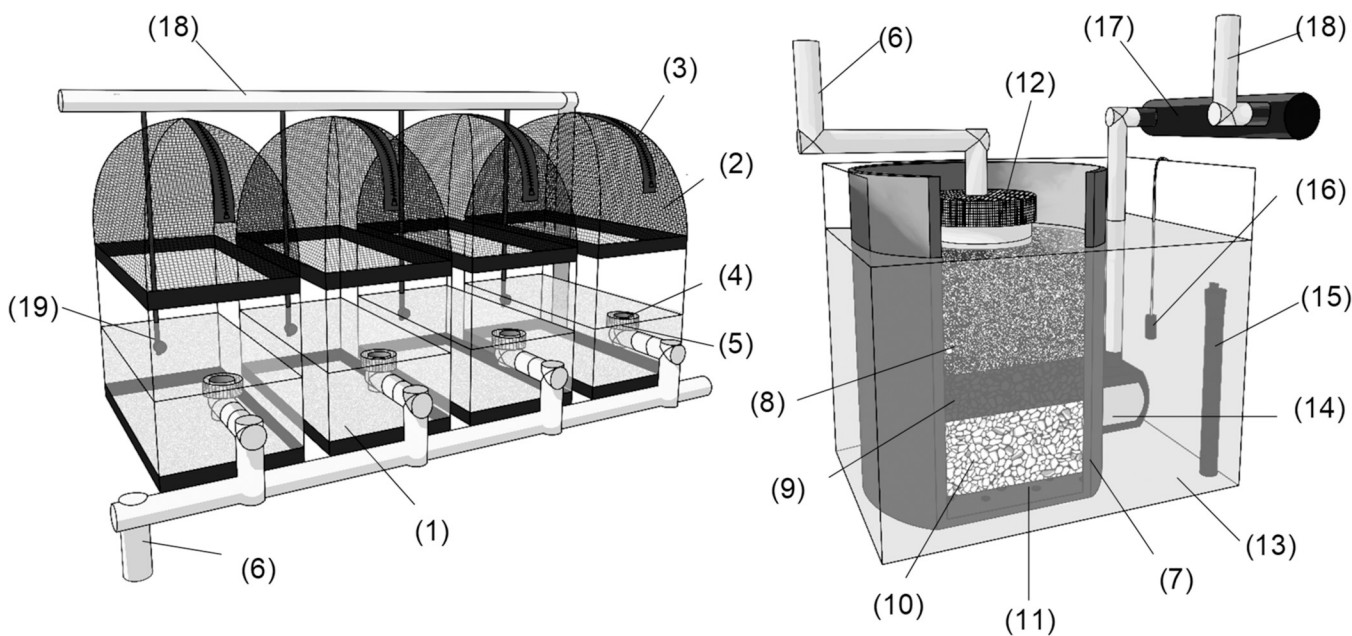

1. Glass aquarium (29 x 16 x 18 cm)
2. Aerial enclosure
3. Clasp lock
4. Outflow pipe to filter chamber
5.Overflow ring
6. Rearing system to filter water pipe
7. 19L container
8. Sand
9. Activated carbon
10. Gravel
11. Drainage
12. Sieve
13. Glass Aquarium (43 x 29 x 38 cm)
14. Water pump
15. Heater
16. Air stone
17. UV sterilizer
18. Water distribution system
19. Flow valve

**Fig 2. System used for the rearing of *Chironomus* sp. "*Florida*".** (Left) Rearing units components. (Right) Filter chamber components.

aerial enclosure was covered with a handmade case of mesh fabric (1 mm pore), held in place with a polyester elastic band and with a straight aperture (15 cm length) on one of the sides sewed and secured with a clasp locker (Fig 2.3). This aperture provided access to the interior of the aquarium without disturbing the enclosed adults.

The overflow system was made by drilling an aperture (2 cm diameter) in one of the short lateral faces of the aquariums, located at 8 cm from the sides and 9 cm from the base. PVC elbows were secured to the openings of each aquarium and used as the overflow water outlets (Fig 2.4). A ring made of plastic canvas (4 cm diameter; 2 cm height) was positioned at the mouth of each outlet to prevent drifting of egg masses, large larvae, and pupae (Fig 2.5). Additional 2-cm PVC tubing was used to connect the aquariums and drain the water to the filter (Fig 2.6).

The filter was made with a 19 L container (Fig 2.7), filled with layers of gravel, activated carbon, and sand (Fig 2.8–2.10). Water was moved by gravity, exiting through holes at the bottom of the container (Fig 2.11). A handmade strainer (nominal sieve opening = 250 μm) was positioned between the filter and the incoming water to prevent the entry of coarse matter, drifting

larvae, and drifting egg masses into the filter (Fig 2.12). This filter was placed inside a glass aquarium (43x29x38 cm) (Fig 2.13) provided with 16 L of dechlorinated tap water, aeration, a water pump, and a heater (Fig 2.14–2.16). The filter was reinforced biologically by adding beneficial bacteria from Imaginarium™ Biological Booster (0.13 ml/L). The water pump inside the aquarium sent the filtered water through a UV filter to decrease the number of bacteria, protozoans, algae, and diatoms (Fig 2.17). From the UV filter, the water was distributed using PVC pipe (2 cm diameter) equipped with irrigation tubing and micro-spray nozzle valves (Fig 2.18–2.19).

## Culture conditions

Each culture aquarium was provided with 4 L of dechlorinated tap water. Water temperature was maintained at 27±1˚C, which is an average temperature that we have observed for lowland streams in Puerto Rico. Air temperature was maintained at 25±1˚C by using a 250-watt ceramic heat lamp, simulating the range of average atmospheric temperature reported for the island (24 to 29˚C [27]). Both temperatures were regulated with digital thermostats. Constant aeration in the filter chamber provided oxygenated water to culture aquariums. A photoperiod of 12:12 dark/light was maintained using 10-watt LED lights with a temperature colour of 6400 K.

## Substrate and diet

The substrate we use for culturing the larval *Chironomus sp.* *"Florida"* was based on the methodology given by Batac-Catalan and White [28]. A key difference in methodology is that instead of shredded paper towels soaked in acetone and boiling water, commercial coffee filters made of oxygen whitened paper was used. The bottom of each aquarium was covered with a 2-cm layer of this material. Larvae were fed with a stock suspension made of Tetramin® fish flakes with tap water. The food suspension was prepared in large batches, separated into individual 1.5-ml Eppendorf tubes™, and stored at -40˚C until use. The larvae in each tank were fed at a concentration of 2 mg/larva (~400 mg/aquarium) of Tetramin® daily.

## Reproduction

Since *Chironomus* sp. *"Florida"* is a species that requires swarms to mate, adults were left in the culture aquariums after emergence. The culture aquarium enclosure provided enough space for adults to conglomerate and for males to start swarming behaviours at the beginning and end of the photoperiod. Because males must be more abundant than females for mating to occur, in some instances adults from all culture aquariums were collected and placed in a single aquarium to obtain egg masses. Adults were collected using 50-ml Falcon® centrifuge tubes and a flashlight to attract them into the container. The same method was used to release them into the recipient aquarium.

## Culture maintenance

From Monday to Friday, each canvas ring was removed and cleaned under tap water. Floating dead adults and exuviae were left to move down through the overflow outlets for 15 minutes. Any decaying material was brushed and removed. The filter strainer was washed under tap water after cleaning the aquariums to remove all the accumulated material. One to two times a week, 50% of the water volume in the filter aquarium was siphoned and replaced with dechlorinated tap water adjusted at the culture temperature. This helped to compensate for water evaporation. Additional cleaning was provided if signs of stress were observed (e.g., larvae

breathing at surface), bad odours were perceived, or overgrowth of bacteria or protozoa was detected. In those cases, all the water of the filter aquarium was siphoned and replaced as many times as needed to recover balance in the system. Every 3 months the sand, activated carbon, and gravel in the filter were rinsed to prevent clogging. Because larval substrate decreases over time, new shredded coffee filter was added every 3 to 4 weeks as needed.

## Life cycle

Life cycle and growth were described following modifications of the methodologies presented by Zilli [29]. Briefly, we collected 45 egg masses from the laboratory colony in a period of 3 days. The eggs in each mass were quantified and incubated at 27˚C until hatching. The hatching time was recorded, and the unhatched eggs were quantified under the microscope. Newly hatched larvae from 11 egg masses and with the same age were randomly separated into six 1000-ml beakers provided with oxygenated water and 2 cm of shredded paper. These larvae were incubated at 27˚C and fed daily as described in the *Diet* section above.

Three of the six beakers were selected to determine larval growth and larval stage. Ten larvae were collected daily from each beaker and fixed in 70% ethanol. The head capsule width and total body length of each larva were measured under a dissecting microscope. Larval growth through time was modelled by relating larval body size and time, fitting the result to the best polynomic curve. Larval instars were identified by obtaining the relationship between head capsule width and total body length following the Dyar proportions [30]. This method was also used to determine the growth rate between instars. The larvae in the remaining three beakers were left to reach adulthood and observed daily to record pupae presence, adult emergence, and daily adult sex distribution. Life cycle parameters such as the time to egg hatching, egg mass hatching efficiency, mean duration of each instar, immature development time, and minimum generation time were determined.

## Effect of feeding and temperature on larval emergence

Each variable was evaluated using recently hatched larvae from five egg masses, segregated individually into sets of 25 small containers (5 ml) per treatment. Each container was provided with 3 ml of dechlorinated tap water, and we did not aerate the containers, assuming good oxygen diffusion between the water and the atmosphere due to the low water volume. To evaluate the effect of feeding, we fed four larval sets either 0.5, 1, 2, or 3 mg of TetraMin® suspension daily. Five additional larvae per the first three food concentration were grown in a different setting using 20 ml of water instead of three. This was done to see if the experiment could be reproduced at higher water volumes obtaining the same results.

The same setting used in the feeding experiment was used to evaluate the effect of temperature: six larval sets were exposed to either 15, 18, 20, 25, 29, or 35˚C. In both experiments, water was exchanged daily to decrease water quality deterioration. The larvae were maintained under experimental conditions until adulthood, when time to reach emergence was recorded for each treatment.

## Larval biomass

Larvae from different stages and sizes were collected from the colonies and euthanized in water with formalin (5%). Each larva was measured under a dissection microscope from the anterior edge of the head capsule to the posterior end of the abdomen. Individual larvae were transferred to a pre-weighed 2-$cm^2$ aluminium sheet, dried at 60˚C for 24 hours, and weighed in an analytical scale (±0.0001 mg). Larval dry body mass was modelled as a function of larval body length using the power model.

## Statistical analysis

Statistical analyses and models were obtained with the assistance of the statistical software GraphPad PRISM 7. Treatments were compared using analysis of variance (One-way ANOVA), and Tukey's post-hoc test was applied to identify differences among individual treatments.

## Results

*Chironomus* sp. *Florida* egg masses had on average 236.98 (n = 45; σ = 26.87) oval eggs that turn brownish when fertilized. Embryos were visible 12 hours after being laid and developed completely in less than 24 hours when incubated at 27˚C. Hatching efficiency of egg masses was 95.56% (n = 45), and for viable egg masses hatching efficiency of eggs was 98.67% (n = 43). After hatching, the larvae stayed in the gelatinous mass and fed on it for 5 to 8 hours before turning planktonic. Larvae passed through four stages or larval instars that were easily identified by morphology or by relating the head width with the body length (Fig 3A). A summary of the values for head width, body length, stage duration, and morphological appearance is presented in Table 1. The average larval developmental time was 11 days (n = 899, σ = 1.31), with a minimum and maximum of 8 and 14 days.

The pupal stage developed in hours, producing a pharate adult that emerged in less than 24 hours. Emergence took place between day 10 and day 15 from egg deposition (n = 899; $X$ = 13.07 σ = 1.31): males appeared between day 10 and day 13 (n = 309, $X$ = 11.52 σ = 0.64); females between day 12 and day 15 (n = 590, $X$ = 13.88, σ = 0.70). With this, the average

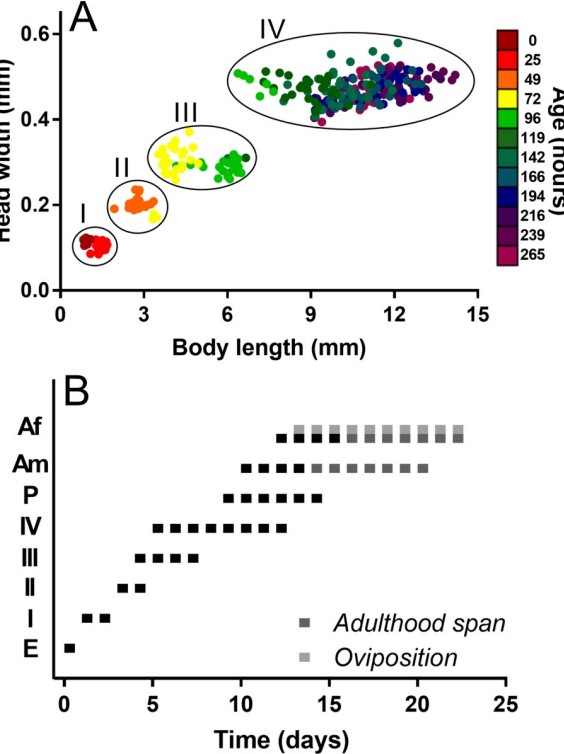

**Fig 3. *Chironomus* sp. "*Florida*" stages.** (A) Larval stages (I to IV) based on the relation between the head capsule width and body length. (B). Average duration and overlap between stages. Abbreviations: E: embryo; I-IV: larval instars I to IV; P: pupa; Am: adult male; Af: adult female. Adult presence without occurring emergence or adult span is represented in dark grey. Oviposition period is represented by light grey.

**Table 1. Stage description, life cycle parameters, larval size ranges and larval growth rates of *Chironomus* sp. "*Florida*".**

| Stage | Description | Life cycle parameters | | Larval size ranges | | Larval growth rates | |
|---|---|---|---|---|---|---|---|
| | | Day of appearance | Average duration (day) | Head width range (mm) | Body length range (mm) | Head | Body |
| **Embryo** | Brownish and oval, secured in gelatinous matrix with other embryos. | 0 | <1 | | | | |
| **Larvae** | | | | | | | |
| Instar I | Clear and planktonic | 1 | 1.71±0.49 | 0.09–0.12 | 0.83–1.68 | 2.02 | 1.50 |
| Instar II | Whitish coloration | 3 | 3.14±0.29 | 0.17–0.24 | 1.94–3.47 | 2.13 | 2.22 |
| Instar III | Pink coloration | 4 | 4.55±0.63 | 0.23–0.37 | 3.52–6.67 | 2.18 | 5.74 |
| Instar IV | Bright red larvae, green colouration can be observed in the thoracic capsule when ready to emerge. Females larger than males. | 5 | 8.90±2.10 | 0.39–0.58 | 6.38–14.19 | | |
| **Pupa** | | | | | | | |
| Male | Bright red or green with evident thoracic horns. Swim actively before emergence. Females bigger than males. | 9 | 1 | | | | |
| Female | | 11 | 1 | | | | |
| **Adult** | | | | | | | |
| Male | Bright green. Male with plumose antenna and slender abdomen. | 10 | 3.63±1.47 | | | | |
| Female | Females with simple antenna, and brownish and broad abdomen | 12 | 5±1.45 | | | | |

duration of the immature stage was 13 days (n = 899, σ = 1.31), with a minimum and a maximum of 9 and 14 days. In high humidity environments (86% humidity) males survived an average of four days (n = 24; σ = 1.47) after emerging, while females survived an average of six days (n = 20; σ = 1.45). Oviposition was observed from day 13 to day 18, producing a new generation from day 13. The complete overlap and duration of stages is illustrated in Fig 3B.

Animals that were fed at a food concentration of 0.5 mg/larva/day presented shorter adult emergence times than those fed at a higher concentration (Fig 4A). These results were not reproducible at a higher water volume since inverse results were obtained, in that the larvae fed at 2 mg/larva/day were the ones that presented shorter emergence times (Fig 4B). Larvae food deprivation was observed in the 0.5, 1, and 2 mg/larva/day treatments for both water volumes. Deprivation occurred first at low food concentrations and later at high food concentrations (Table 2).

Rearing larvae at temperatures as low as 15°C lengthened the time for adult emergence (Fig 4C). As the rearing temperature is increased, emergence time shortened. According to our model, larvae reared above 29°C are expected to begin to show an increase in the time of emergence. Both, feeding concentration and temperature presented a polynomial relationship with emergence time (Fig 4).

Larval growth at 27°C from the first instar larva to the prepupae stage was best represented by using a polynomial model (Fig 5A). This growth model showed a rapid increase in larval size until reaching the ninth day, when growth started to decrease. Growth rates between instars increased slightly through time, with the last stage being the one with the highest growth rate value (Table 1). The relationship between body length and dry mass was best explained using the power model, rather than linear or quadratic models (Fig 5B).

## Discussion

Our method was the result of 3 years of observations on the suitability of different methodologies, including modified versions of Biever [31], New [32], and Batac-Catalan and White [28]. The preference of collecting eggs for establishing new colonies and studies is common and

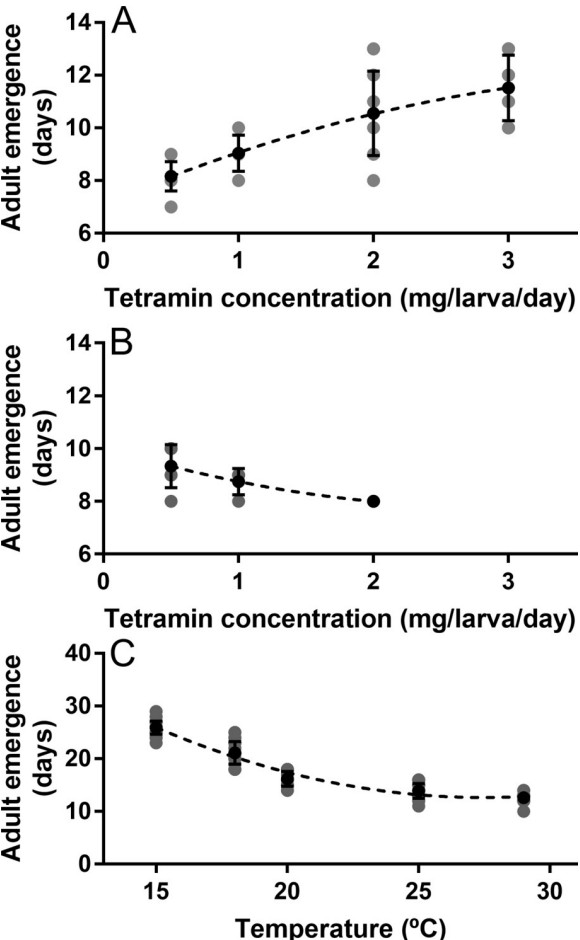

**Fig 4. Effect of feeding and temperature on adult emergence.** (A) Emergence curve obtained from larvae fed at four food concentrations in a water volume of 3 ml. (B) Emergence curve generated from larvae fed at three food concentrations in a water volume of 20 ml. (C) Emergence curve obtained from larvae reared at five different temperatures. Average emergence times and standard deviations are represented in black; individual emergence values are represented in grey. Each Polynomial model can be expressed as (A) $y = -0.24x^2 + 2.17x + 7.12$, $R^2 = 0.60$; (B) $y = 0.28x^2 - 1.58 + 10.05$, $R^2 = 0.54$; (C) $y = 0.08x^2 - 4.69x + 77.13$, $R^2 = 0.98$.

favoured by various authors (e.g., [14, 29, 33–35]). With the proposed recirculated system, egg masses can be used to produce and maintain robust colonies without effort for a period longer than 9 months. Although recirculated systems are not commonly used for rearing and maintaining chironomids owing to larval drifting problems, it has the advantage of producing a constant oxygenated, clean, and biologically uncontaminated environment that reduces maintenance time. In our experience, rearing methods such as the ones presented by Biever [31],

**Table 2. Time to food deprivation at different water volumes for each feeding treatment.**

| Food concentration (mg/larva/day) | Time to food deprivation at a water volume of 3ml (days) | Time to food deprivation at a water volume of 20ml (days) |
|---|---|---|
| 0.5 | 3 | 2 |
| 1 | 4 | 4 |
| 2 | 6 | 6 |
| 3 | Not observed | - |

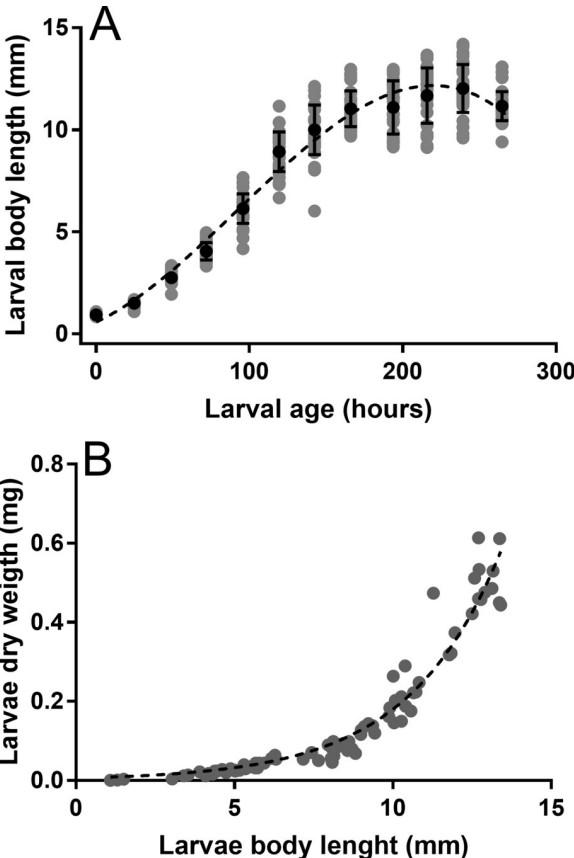

**Fig 5. Growth and dry biomass equivalence of the larval *Chironomus* sp. "*Florida*".** (A) Relationship of body size and age. (B) Relationship between body size and biomass. Average values and respective standard deviations are represented in black while individual measurements are represented in grey. The polynomial and power regressions can be expressed respectively as (A) $y = -1.37e^{-6} + x^3 + 0.24x^2 + 0.35e^{-3} x + 0.34$ $R^2 = 0.94$; (B) $y = 0.006e^{(0.34^*x)}$, $R^2 = 0.94$.

Batac-Catalan and White [28], and Ingersoll [13] do not provide long-term stability, and colonies start to present problems with overgrowth of microorganisms with subsequent anoxia, something that is rarely observed with recirculated systems.

We exchanged the material used as substrate by Batac-Catalan and White [28] since shredded coffee filters offer a fibrelike medium that is easy to prepare. This media has a high acceptance by larvae and is used as burrowing and case construction material. The only disadvantage of this substrate is that it starts degrading into a fine material, hindering its posterior reuse by the larvae.

Tetramin® flakes have often been used to feed larvae [33, 36–38]. Although other rearing methodologies have used a range of feeding sources (e.g., vegetative organic matter, animal chow, manure, algae, and yeast), the use of Tetramin® has been shown to provide larvae with essential amino acids that promote enhanced growth rates in laboratory cultures [39]. It is important to mention that our proposed feeding concentration is based on observing how much food the larvae were able to ingest daily. Therefore, further studies are necessary to determine possible effects of the feeding concentrations on reproduction, survivability, and biomass. As in other rearing methodologies, adults were not fed in our method. This approach is based on the idea that adults are short lived and mostly rely on energy obtained during the immature stages to reproduce [40].

Our recommended method of aerial reproduction is the optimal way to achieve mating of adults in this species. Other methods, such as confined reproduction chambers, cannot be used since adults need space to swarm to achieve mating. This aerial method has been used elsewhere to keep producing new laboratory generations of species with swarming mating behaviours [15, 41, 42]. This method has provided us with a productivity of more than 15 egg masses/day per rearing unit once colonies are established. The recommended environmental conditions (27˚C; photoperiod 12light:12dark) are similar to those observed in the wild environment of the species and are in accordance with other methodologies from tropical regions [43]. Nevertheless, it is important to highlight that these conditions are not necessarily the optimal ones for obtaining high biomass and progeny outputs.

## Life cycle

The life cycle of most species in the genus *Chironomus* is similar and is characterized by an embryo enclosed in a gelatinous mass, four larval stages, a pupa, and an adult [44]. Egg masses of *Chironomus* sp. "*Florida*" are small (about 236 eggs) relative to other species. Common tropical species of *Chironomus* laid over 350 eggs/mass (e.g., *C. calligraphus*, 369–374 eggs/mass; [14]), reaching up to 3,300 in some species [45, 46]. The low number that we found is consistent with the medium body size of adult *Chironomus* sp. "*Florida*", as small species are expected to lay fewer eggs per mass [45].

Despite the smaller number of eggs in our species, it exhibits an excellent reproductive output based on the number of hatching eggs (98.67%). This high value is in accordance with other species such as *C. riparius* (97.78%) [45] and *C. pulcher* (99%) [47]. Hatching starts in less than 24 hours, contrasting with *C. calligraphus* (3 days at 21.8 ±3.2˚C) [29], *C. xanthus* (1.25 days at 30˚C) [48], and *C. riparius* (2–6 days at 26˚C) [45]. This faster embryo development could represent an adaptive advantage to increase fitness, as this species inhabits temporary habitats. The typical egg mass feeding and planktonic behaviour after hatching are preserved in this species, a topic that has been observed and discussed elsewhere [14, 36, 49].

*Chironomus* sp. "*Florida*" has a short life cycle, with a minimum duration of the immature stage ($D_{min}$) of only 9 days at 27˚C. This is comparable to other tropical species that have presented similar $D_{min}$ values in a range of the temperatures of 24–30˚C such as *C. strenzkei* ($D_{min}$ = 10 days [50]), *C. sancticaroli* ($D_{min}$ = 15 days [51]), *C. xanthus* ($D_{min}$ = 13 days [36]), *C. calligraphus* ($D_{min}$ = 11 days [14]), *C. crassiforceps* ($D_{min}$ = 10 days [43]), and *C. ramosus* ($D_{min}$ = 15 days [52]). As noted for egg hatchability, this low $D_{min}$ could be explained as an advantage for inhabiting temporary habitats.

We used the Dyar proportions to identify the four larval stages, determine growth rate, and obtain stage durability values. This method has been recommended as head measurement ranges are distinctive through each larval stage, something that is not observed with body measurement ranges [29, 36, 53]. In general, head size ranges and growth rates reported in this study for each larval stage are higher than those reported for other tropical species (e.g., *C. crassiforceps* [43], *C. sancticaroli* [54], and *C. xanthus* [36]). The average duration of each larval stage is dissimilar from the other stages, increasing every time the larva moves to its next phase. The last larval stage is therefore longer than the previous ones. This situation has been reported for other species, and it has been explained by Tokeshi [40] as a way to acquire the necessary energy for swarming activity in adulthood.

The larval growth model for *Chironomus* sp. "*Florida*" presents an increment in body size during the initial stages until reaching a period of slow growth at the final larval stage. This kind of growth has been reported in species under the *Chironomus* group [29], but also in species of other genera, such as in the *Goeldichironomus* group [33, 34]. In this growth model, we

can observe two unusual behaviours: 1) as larvae age, the variability in larval body size increases; 2) as the larvae in the cohort reach a maximum in growth, a decrease in body size is observed afterward. In the case of the first observation, the dissimilarities in body size could be produced by sexual dimorphism [55] and by intraspecific competition for food and space [56]. For the second observation, a decrease in body size could be explained by the appearance of the prepupae, a stage characterized by a decrease in thoracic length due to increased swelling. It is important to note that the growth models presented by Corbi and Trivinho-Strixino [33], Zilli [29], and Zilli [34] do not show this decrease. Our observation may result from the use of a different statistical model in our study.

It is important to take into consideration that males and females were not separated to produce the growth curve presented in Fig 5B. This detail is something that this study and other studies have ignored when executing this kind of model because of the complexity of identifying the sex at the larval stage By constructing models by sex, we could reduce variability and develop more precise growth curves that eliminate the effect of sexual dimorphism.

The pupal and adult stages in chironomids are in general short, ranging from hours to a few days [44, 57]. The pupa of *Chironomus* sp. *"Florida"* is not an exception, presenting a developmental time of hours (less than 24) similar to *C. crassicaudatus* (22–27 hours at 27˚C [53]) and *C. riparius* (24 hours at 26˚C [45]). The longevity of adults in this species was unexpectedly longer (more than 2 to 3 days), something rarely reported for tropical species. Our longevity values could be the consequence of our experimental conditions provoking restricted flight and reduced dehydration (due to the way we contained the animals). This is confirmed by studies in chironomids and other insects that have demonstrated how decreasing flight and dehydration increase survivability [58, 59].

## Effect of temperature and feeding on adult emergence

Temperature and food availability are two variables known to control larval development in chironomids [38, 41, 45]. Regarding adult emergence, it has been shown that both variables interfere in the duration of this process [60]. Typically, adult emergence time is shortened by an increasing rearing temperature, but the emergence time may slightly increase again if the temperature rises above 30˚C. This decreasing adult emergence time behaviour is present in *Chironomus* sp. *"Florida"* This species, however, presents a low tolerance to high temperatures since no survivorship was achieved at 35˚C. This was unexpected given that this species inhabits the tropics and other species of chironomids can survive in temperatures above 35˚C [61].

Food deprivation is known to have several effects on chironomids, from changes in fitness to changes in life cycle parameters [38, 62]. The results obtained at low water volume were surprising since an increase in emergence time was expected. Notably, these data were not reproducible at high water volumes, a finding that led us to conclude that the observed behaviour at small water volumes was caused by oxygen depletion related to bacterial overgrowth due to excess of uneaten food. Confirmatory evidence was observed after each water change in the treatments where food concentrations were greater than 1 mg/larva/day. In this case, larvae were observed at the water surface producing constant breathing movements. This behaviour was not observed in experiments with large water volumes and high food concentrations where larvae were observed at the bottom of the container with no observable breathing behaviour. Although other feeding experiments have mentioned that larvae fulfil nutritional demands at concentrations as low as 0.12 mg/larva/day [38], here we observed that, in a species that grows fast, concentrations of 2 mg/larva/day are required to avoid food deprivation. Nevertheless, in order to achieve faster adult emergence times, it is necessary to avoid unacceptable water conditions due to oxygen depletion and waste production.

In summary, this study showed that a native non-biting species of the *Chironomus* genus could be easily reared under laboratory conditions in Puerto Rico. The native species *Chironomus* sp. *"Florida"* is ideal for this purpose, since it is easy to obtain in the field, easy to maintain under laboratory conditions, and has a short life cycle. The use of a native species model for freshwater toxicological evaluations provides a better understanding of how contaminants affect tropical ecosystems.

## Supporting information

**S1 File. Original data from all experiments.**
(XLSX)

## Acknowledgments

We thank Taissae Sánchez Medina for her advice and general support in the maintenance of colonies and experiments. We are also grateful to Gloria Ortiz for helping us obtain some of the materials and instruments required for this study.

## Author Contributions

**Conceptualization:** Roberto Reyes-Maldonado, Bruno Marie, Alonso Ramírez.

**Formal analysis:** Roberto Reyes-Maldonado.

**Funding acquisition:** Bruno Marie, Alonso Ramírez.

**Investigation:** Roberto Reyes-Maldonado.

**Methodology:** Roberto Reyes-Maldonado.

**Project administration:** Roberto Reyes-Maldonado, Bruno Marie, Alonso Ramírez.

**Resources:** Roberto Reyes-Maldonado, Bruno Marie, Alonso Ramírez.

**Supervision:** Bruno Marie, Alonso Ramírez.

**Validation:** Roberto Reyes-Maldonado.

**Visualization:** Roberto Reyes-Maldonado.

**Writing – original draft:** Roberto Reyes-Maldonado.

**Writing – review & editing:** Roberto Reyes-Maldonado, Bruno Marie, Alonso Ramírez.

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
