## [Decision Letter · Decision Letter 0]

5 Jan 2021

PONE-D-20-34738

Rearing methods and life cycle characteristics of Chironomus sp. Florida (Chironomidae: Diptera): a rapid-developing species for laboratory studies

PLOS ONE

Dear Dr. Reyes-Maldonado,

Thank you for submitting your manuscript to PLOS ONE. After careful consideration, we feel that it has merit but does not fully meet PLOS ONE’s publication criteria as it currently stands. Therefore, we invite you to submit a revised version of the manuscript that addresses the points raised during the review process.

Given that Methods sections are typically written in the past tense, I would ask that you make the changes necessary to fit this standard style.  In addition, I ask that you consider the comments from Reviewer 2 regarding reducing the overall length of the manuscript. Lastly, delete the "et al." in line 320 for "...New et al. [32]....".

We look forward to receiving your revised manuscript.

Kind regards,

J Joe Hull, Ph.D.

Academic Editor

PLOS ONE

Journal Requirements:

2.) PLOS requires an ORCID iD for the corresponding author in Editorial Manager on papers submitted after December 6th, 2016. Please ensure that you have an ORCID iD and that it is validated in Editorial Manager. To do this, go to ‘Update my Information’ (in the upper left-hand corner of the main menu), and click on the Fetch/Validate link next to the ORCID field. This will take you to the ORCID site and allow you to create a new iD or authenticate a pre-existing iD in Editorial Manager. Please see the following video for instructions on linking an ORCID iD to your Editorial Manager account: https://www.youtube.com/watch?v=_xcclfuvtxQ

Reviewers' comments:

Reviewer's Responses to Questions

**Comments to the Author**

1. Is the manuscript technically sound, and do the data support the conclusions?

Reviewer #1: Yes

Reviewer #2: Yes

2. Has the statistical analysis been performed appropriately and rigorously? 

Reviewer #1: Yes

Reviewer #2: Yes

3. Have the authors made all data underlying the findings in their manuscript fully available?

Reviewer #1: Yes

Reviewer #2: Yes

4. Is the manuscript presented in an intelligible fashion and written in standard English?

Reviewer #1: Yes

Reviewer #2: Yes

5. Review Comments to the Author

Reviewer #1: The manuscript is technically sound, and the data support the conclusions. It describes a technically sound study about rearing techniques of the non-biting midge Chironomus sp. "Florida". Experiments were conducted with appropriate controls and sampling size. Conclusions are appropriate.

Statistical analysis were performed appropriately and described well.

As stated by the authors, all data underlying the findings are fully available.

The manuscript is written well, the language is clear and correct.

Minor Review: Although the manuscript is well written, covers an interesting experiment/technology with all nessecary details, and is of interest for the field, I would recommend that it could be shortened without loosing information on details.

Reviewer #2: This is a very important and interesting manuscript describing rearing methods and life cycle characteristics of a very neglected group of aquatic insects. The articles is well-written and bring lots of relevant information to the field. This manuscript should be accepted as it is.

6. PLOS authors have the option to publish the peer review history of their article (what does this mean?). If published, this will include your full peer review and any attached files.

Reviewer #1: No

Reviewer #2: No

---

## [Author Response · Author response to Decision Letter 0]

3 Feb 2021

Following Reviewer #1 recommendation, we have also edited the text, reducing the size as recommended by simplifying sentences, paragraphs and subheadings.

---

## [Editor Report · Decision Letter 1]

8 Feb 2021

Rearing methods and life cycle characteristics of Chironomus sp. Florida (Chironomidae: Diptera): a rapid-developing species for laboratory studies

PONE-D-20-34738R1

Dear Dr. Reyes-Maldonado,

We’re pleased to inform you that your manuscript has been judged scientifically suitable for publication and will be formally accepted for publication once it meets all outstanding technical requirements.

Kind regards,

J Joe Hull, Ph.D.

Academic Editor

PLOS ONE
---

## [Editor Report · Acceptance letter]

11 Feb 2021

PONE-D-20-34738R1 

Rearing methods and life cycle characteristics of *Chironomus* sp. Florida (Chironomidae: Diptera): a rapid-developing species for laboratory studies 

Dear Dr. Reyes-Maldonado:

I'm pleased to inform you that your manuscript has been deemed suitable for publication in PLOS ONE. Congratulations! Your manuscript is now with our production department. 

Kind regards, 

on behalf of

Dr. J Joe Hull 

Academic Editor

PLOS ONE